## [Decision Letter · Decision Letter 0]

20 Oct 2025

The chromatin remodeling factor Arp9 modulates multidrug-resistance and plays a key role in aflatoxins biosynthesis under mammalian-physiological-temperature in Aspergillus flavus

PLOS Pathogens

Dear Dr. Zhuang,

Thank you for submitting your manuscript to PLOS Pathogens. After careful consideration, we feel that it has merit but does not fully meet PLOS Pathogens's publication criteria as it currently stands. Therefore, we invite you to submit a revised version of the manuscript that addresses the points raised during the review process.

Please submit your revised manuscript within 60 days Dec 19 2025 11:59PM. If you will need more time than this to complete your revisions, please reply to this message or contact the journal office at plospathogens@plos.org. Please include the following items when submitting your revised manuscript:

We look forward to receiving your revised manuscript.

Kind regards,

Haoping Liu

Academic Editor

PLOS Pathogens

Alex Andrianopoulos

Section Editor

Editor-in-Chief

PLOS Pathogens

Editor-in-Chief

PLOS Pathogens

orcid.org/0000-0002-7699-2064

**Journal Requirements:**

At this stage, the following Authors/Authors require contributions: Zhenhong Zhuang. Please ensure that the full contributions of each author are acknowledged in the "Add/Edit/Remove Authors" section of our submission form.

Potential Copyright Issues:

- Please confirm (a) that you are the photographer of Figures 3A, 3C, 4A, and 4D., or (b) provide written permission from the photographer to publish the photo(s) under our CC BY 4.0 license.

5) Please ensure that the funders and grant numbers match between the Financial Disclosure field and the Funding Information tab in your submission form. Note that the funders must be provided in the same order in both places as well.

**Reviewers' Comments:**

Reviewer's Responses to Questions

**Part I - Summary**

Reviewer #1: The authors have addressed all my queries satisfactorily. The revised manuscript is much improved. Hence, I think the manuscript can be now accepted for publication.

Reviewer #2: I have reviewed this manuscript previously and have comments for this part. In general, novelty of this study is good.

**Part II – Major Issues: Key Experiments Required for Acceptance**

Reviewer #1: (No Response)

Reviewer #2: In general, authors have responded all comments point by point. Authors really tried to make efforts to improve manuscript quality such as they did swi1, sfh1, and snf2 deletion mutants and found they all exhibited increased resistance to VOR compared to the WT which is consistent with that of dArp9, suggesting Arp9 and its putative complex partners had similar functions to them but we still could not conclude they are partners since there are no Co-IP biochemistry data or other direct evidence data.

However, function of Arp9 as a chromatin remodeling factor still is not very solid only based on transcriptome analysis, qRT-PCR, ChART-qPCR. Why the deletion of Arp9 exacerbated their resistance to VOR and what is molecular mechanism? No working model to be understood for the function of Arp9. In addition, there are still have some confused conclusions such as

as they demonstrated that swi1, sfh1, and snf2 deletion mutants exhibited increased resistance to VOR compared to the WT (Fig. 8C). However, these mutants displayed the opposite phenotype to AMB compared to the ΔArp9? how they have opposite functions for their targeted genes transcription?

**Part III – Minor Issues: Editorial and Data Presentation Modifications**

Reviewer #1: (No Response)

Reviewer #2: minor comments:

in the title: Arp9 modulates multidrug-resistance. However, AMB has an opposite effect with that of VOR for Arp9 mutants

Figure 3G. what is indication for red cycle of H&E staining for observation

PLOS authors have the option to publish the peer review history of their article (what does this mean? ). If published, this will include your full peer review and any attached files.

**Do you want your identity to be public for this peer review?** For information about this choice, including consent withdrawal, please see our Privacy Policy .

Reviewer #1: No

Reviewer #2: No

**Figure resubmission:**

**Reproducibility:**



---

## [Editor Report · Decision Letter 1]

7 Jan 2026

PPATHOGENS-D-25-02077R1

The chromatin remodeling factor Arp9 modulates drug-resistance and plays a key role in aflatoxins biosynthesis under mammalian-physiological-temperature in Aspergillus flavus

PLOS Pathogens

Dear Dr. Zhuang,

Thank you for submitting your manuscript to PLOS Pathogens. After careful consideration, we feel that it has merit but does not fully meet PLOS Pathogens's publication criteria as it currently stands. Therefore, we invite you to submit a revised version of the manuscript that addresses the points raised during the review process.

1. While you performed the suggested experiments to address questions regarding the arp9 phenotype in the context of the SWI/SNF and RSC complexes, you did not clearly distinguish between these two complexes in the writing. Line 108 should provide more information on Arp9, such as: "Arp9 (actin-related protein 9) is a member of both the SWI/SNF and RSC chromatin remodeling complexes in fungi (refs)." In the results section that describes Fig. 8S and Fig. 8C, please clarify: "Sfh1 (Snf5 homolog 1) is a key component of the yeast RSC complex, but not part of the SWI/SNF complex (ref). Swp82 is a component of the SWI/SNF complex (ref)."

2. Please expand the abbreviation ChART-qPCR (Chromatin Accessibility by Real-Time PCR) upon first use in the text and define relative CAI (chromatin accessibility index) in the Fig. 5E legend. Line 370 should read "qRT-PCR and ChART-qPCR." Typically, ChART-qPCR uses specific primer pairs tiling across a gene of interest. It is not clear how you determined the location of just one primer pair for each promoter examined in Fig. 5E. Additionally, Line 672 in the methods section states "ChART-rtPCR was performed to analyze gene expression levels," which is misleading, as ChART-qPCR measures chromatin accessibility, not gene expression.

3. Lines 441-448: The new Fig. S9 presents data that are not MNase-seq or ATAC-seq and therefore cannot provide direct information regarding nucleosome sliding. This content would be better suited for the discussion section.

4. The abstract should be updated to reflect the revisions. The writing throughout the entire manuscript should be reviewed to improve clarity and expression. For example, Lines 122-123 use the phrase "individual level," which is unclear in this context and should be clarified.

We look forward to receiving your revised manuscript.

Kind regards,

Haoping Liu

Academic Editor

PLOS Pathogens

Michal Olszewski

Section Editor

PLOS Pathogens

Sumita Bhaduri-McIntosh

Editor-in-Chief

PLOS Pathogens

orcid.org/0000-0003-2946-9497

Michael Malim

Editor-in-Chief

PLOS Pathogens

orcid.org/0000-0002-7699-2064

**Journal Requirements:**

1) In the online submission form, you indicated that your data will be submitted to a repository upon acceptance. We strongly recommend all authors deposit their data before acceptance, as the process can be lengthy and hold up publication timelines. Please note that, though access restrictions are acceptable now, your entire minimal dataset will need to be made freely accessible if your manuscript is accepted for publication. This policy applies to all data except where public deposition would breach compliance with the protocol approved by your research ethics board. If you are unable to adhere to our open data policy, please kindly revise your statement to explain your reasoning and we will seek the editor's input on an exemption.

2) Please amend your detailed Financial Disclosure statement. This is published with the article. It must therefore be completed in full sentences and contain the exact wording you wish to be published.

2) If any authors received a salary from any of your funders, please state which authors and which funders..

**Reviewers' Comments:**

**Figure resubmission:**
---

## [Editor Report · Decision Letter 2]

11 Feb 2026

PPATHOGENS-D-25-02077R2

The chromatin remodeling factor Arp9 modulates drug-resistance and plays a key role in aflatoxins biosynthesis under mammalian-physiological-temperature in Aspergillus flavus

PLOS Pathogens

Dear Dr. Zhuang,

Thank you for submitting your manuscript to PLOS Pathogens. After careful consideration, we feel that it has merit but does not fully meet PLOS Pathogens' publication criteria as it currently stands. Therefore, we invite you to submit a revised manuscript that addresses the points raised during the review process. Please ensure that the requested corrections are finalized to avoid additional rounds of review.

We look forward to receiving your revised manuscript.

Kind regards,

Haoping Liu

Academic Editor

PLOS Pathogens

Michal Olszewski

Section Editor

PLOS Pathogens

Sumita Bhaduri-McIntosh

Editor-in-Chief

PLOS Pathogens

orcid.org/0000-0003-2946-9497

Michael Malim

Editor-in-Chief

PLOS Pathogens

orcid.org/0000-0002-7699-2064

**Additional Editor Comments:**

The revision inadequately addresses the Abstract. The authors do not seem to agree that the methods they used to assess promoter accessibility are outdated and non-conclusive, yet they still included them in the abstract.

1. Replace “Transcriptome analysis, qRT-PCR, ChART-qPCR (Chromatin Accessibility by Real-Time PCR) and HPLC assay determined” with “We show”.

2. Remove “Micrococcal nuclease (MNase) digestion assays revealed that Arp9 deficiency impairs the rate of nucleosome release from chromatin, suggesting that Arp9 may play a critical role in maintaining chromatin accessibility.”

Here is a revised abstract with additional changes in red:

Aspergillus flavus is the second most prevalent species of Aspergillus causing invasive aspergillosis, but its treatment has been hindered by the continuous emergence of drug-resistant fungal strains, while the underlying mechanisms remain largely unexplored. In this study, we investigated the role of the chromatin remodeling factor Arp9 in A. flavus drug-resistance. We show that Arp9 up-regulates the chromatin accessibility of the Erg3 and Erg6 promoters, thereby increasing their transcription levels and enhancing ergosterol synthesis. Therefore, the absence of Arp9 enhances A. flavus sensitivity to amphotericin B (AMB). Additionally, by down-regulating the chromatin accessibility of the Erg11A gene promoter, Arp9 deletion decreases its transcription levels and subsequently reduces A. flavus resistance to voriconazole (VOR). Co-immunoprecipitation analysis revealed that Arp9 exists in both SWI/SNF and RSC complex. Drug susceptibility test results indicated that the drug sensitivity response induced by Arp9 may be unique to Arp9, as neither SWP82 of the SWI/SNF nor Sth1 of the RSC is not required. The role of Arp9 in drug-resistance was also confirmed using the Galleria mellonella model. Furthermore, we found that VOR induces aflatoxin B1 (AFB1) biosynthesis in an Arp9-dependent manner at 35°C and 37°C, and the effect is dramatically magnified in the VOR-resistant A. flavus strain. This study demonstrates that Arp9 plays a critical role in regulating fungal drug-resistance in vitro and in vivo and reveals that Arp9 is an important factor in enhancing AFB1 biosynthesis under Mammalian physiological temperatures. This study provides potential new insights for the control of infections caused by filamentous pathogenic fungi.

3. The revision does not address the methodological issues used to conclude promoter accessibility.

Line 690, the ChART-qPCR method referenced publication # 76 (published 1997), but this paper only provides a method for nuclease digestion and DNA extraction. This publication did not use qPCR or name their method ChART-qPCR. A publication that established the ChART-qPCR method is needed if the author followed the published method. The author still have not address or provide experimental evidence for how they selected the primer set used in Figure 5E.

Line 448 -450, change “Furthermore, to investigate… “ to “Furthermore, we performed an MNase digestion assay. “

**Journal Requirements:**

1) In the online submission form, you indicated that your data will be submitted to a repository upon acceptance. We strongly recommend all authors deposit their data before acceptance, as the process can be lengthy and hold up publication timelines. Please note that, though access restrictions are acceptable now, your entire minimal dataset will need to be made freely accessible if your manuscript is accepted for publication. This policy applies to all data except where public deposition would breach compliance with the protocol approved by your research ethics board. If you are unable to adhere to our open data policy, please kindly revise your statement to explain your reasoning and we will seek the editor's input on an exemption.

**Reviewers' Comments:**

**Figure resubmission:**
---

## [Editor Report · Decision Letter 3]

22 Feb 2026

Dear Dr. Zhuang,

We are pleased to inform you that your manuscript 'The chromatin remodeling factor Arp9 modulates drug-resistance and plays a key role in aflatoxins biosynthesis under mammalian-physiological-temperature in Aspergillus flavus' has been provisionally accepted for publication in PLOS Pathogens.

Best regards,

Haoping Liu

Academic Editor

PLOS Pathogens

Michal Olszewski

Section Editor

PLOS Pathogens

Sumita Bhaduri-McIntosh

Editor-in-Chief

PLOS Pathogens

orcid.org/0000-0003-2946-9497

Michael Malim

Editor-in-Chief

PLOS Pathogens

orcid.org/0000-0002-7699-2064
---

## [Editor Report · Acceptance letter]

Dear Dr. Zhuang,

We are delighted to inform you that your manuscript, "The chromatin remodeling factor Arp9 modulates drug-resistance and plays a key role in aflatoxins biosynthesis under mammalian-physiological-temperature in Aspergillus flavus," has been formally accepted for publication in PLOS Pathogens.

Best regards,

Sumita Bhaduri-McIntosh

Editor-in-Chief

PLOS Pathogens

orcid.org/0000-0003-2946-9497

Michael Malim

Editor-in-Chief

PLOS Pathogens

orcid.org/0000-0002-7699-2064